# Performance on Multiple Pilot-Based Grouping Methods in Satellite-Terrestrial Cooperative Wireless Networks

**Jheng-Sian Li [1], Jyh-Horng Wen [2], Chung-Hua Chiang [3] and Chien-Erh Weng [4],***

1  Department of Technology Services, Chunghwa Telecom Co., Taoyuan 326402, Taiwan;
   metlipaper@gmail.com
2  Department of Electrical Engineering, Tunghai University, Taichung 407224, Taiwan; jhwen@thu.edu.tw
3  Network Planning and Optimization Department in Nokia, Taichung 407224, Taiwan;
   preston.chiang@nokia.com
4  Department of Telecommunication Engineering, National Kaohsiung University of Science and Technology,
   Kaohsiung 81157, Taiwan
*  Correspondence: ceweng@nkust.edu.tw

**Abstract:** The grouping method is an efficient transmitting strategy in a spatial diversity system. In this paper, relay-based grouping methods in satellite terrestrial cooperative wireless networks are proposed. The proposed methods focus on finding out the best two signals received from relay stations in a user's neighborhood and use the advantage of space diversity to overcome the effect of channel fading. A grouping method, called the pilot-based grouping method was proposed in our previous work. In order to improve the grouping success rate and the channel capacity, a modified grouping method is proposed. In addition, for a single relay the modified grouping method can achieve better results than the pilot-based grouping method. In the end, several analysis strategies of the grouping method for multimedia broadcast and multicast services in satellite terrestrial cooperative networks are proposed. The simulation results show the performance improvement and the system evaluation for different quality of services in the demanded relay-based cooperative networks. The proposed modified grouping methods can be widespread in any relay-based cooperative networks.

**Keywords:** satellite terrestrial cooperative wireless networks; pilot-based grouping method; relay-based cooperative networks; quality of services

## 1. Introduction

Relay-based cooperative networks are widely studied in many communication systems. In a cellular system [1], the authors analyzed the user-assisted relaying network and numerically evaluated the improvement of user transmission rate in the cooperative network. In [2,3], the authors wanted to find out the optimal relay location to extend cellular coverage. For device-to-device (D2D) cooperation, Cao, Jiang, and Wang proposed a relay mode selection and channel assignment scheme to increase the throughput [4]. The relay density for the D2D network for getting better system gain was also discussed in [5]. In a visible light communication (VLC) system, Kizilirmak, Narmanlioglu, and Uysal evaluated the BER performance with amplify-and-forward or decode-and-forward relaying through power allocation of source and relay terminals [6]. In a satellite communication system, the terrestrial stations can be treated as relays to retransmit the signal from satellite site. Each relay can share their antenna to set up as a virtual multiple antenna system [7–12]. In addition to providing better performance, the STC in the wireless network was proposed in [13,14]. In the commercial applications field, many entrepreneurs operate their satellite-terrestrial communication system in different technologies. In Canada, XM Satellite Radio (XMSR) implements terrestrial stations in dense urban cities to boost the satellite signals for its digital radio service. In the US, Mobile Satellite Ventures (MSV) integrates the satellite and

terrestrial cellular system to offer a satellite telephony system. In Korea, the Satellite Digital Multimedia Broadcasting (S-DMB) system has also started the commercial service since 2006 [15]. S-DMB also uses a terrestrial repeater network for indoor coverage in urban areas.

In view of the foregoing, serval fields of relay-based cooperative networks have been studied. The common concept of these studies is using relays to exploit the spatial gain of the network, but few studies contain relay management as their subject. Therefore, we are interested in how to categorize the relays to realize efficient utility of the network capacity. In [16], the terrestrial stations in the satellite-terrestrial system are separated into several groups. The number of channels are the same for every station to provide data transmission. The stations in the same group will use the same channel to operate cooperatively. In order to obtain the space diversity gain, the user equipment performs the STC decoding process for each group, and all the decoded signals from different channels are combined to recover the original signals. There are several disadvantages in [16]. First, the transmitting power is inefficient, because not all the terrestrial stations are close enough to the user equipment. The further the station is from the user equipment, the more transmitted power should be transmitted. The user equipment has to receive all the signals from each group. If the number of groups increases, the process time of the user equipment also extends.

In order to improve the disadvantage in [16], we had proposed a grouping method to increase the channel capacity in [17]. In this paper, we present a modified grouping method to get better performance than we proposed in [17]. Moreover, the single relay allowed modified grouping method has been proposed to increase the channel capacity. In addition, multimedia broadcast and multicast service (MBMS) is becoming increasingly prevalent in the world today. At the end of this paper, multimedia broadcast requirements are taken into account. Different type of services, such as audio, video, and HD-video, would be assigned different channels for their requests. There are several scenarios for the grouping method to evaluate the system performance. Through the analysis of grouping methods, the evaluation results will show the effects of different service assignments.

The rest of this paper is organized as follow: Section 2 introduces the modified grouping method and single relay allowed grouping method. Section 3 shows the different scenarios for multi-services in the relay-based networks. Finally, the conclusions are presented in Section 4.

## 2. The System Model of Relay-Based Cooperative Networks

### 2.1. Pilot-Based Grouping Method

Figure 1 shows the concept of the relay-based system in [16]. An encoded signal sequence of $[s, P_1, P_2, \ldots, P_N]$ with a code rate of $1/(N + 1)$ is generated by a rate compatible turbo coder in the transmitter of the source node ($N$ is the total number of (frequency) channels). In the encoded sequence, $s$ and $P_i$ ($1 \le i \le N$) stand for a systematic and the $i$-th parity parts, respectively. The received information is assumed to be decoded by the combination of a systematic part and any other parity parts. The relays are assumed to be split into groups of $q$ relays, and there are $N - 1$ relay groups each one consisting of $q$ relay elements. Then the source node transmits the sequence $[s, P_1]$ to the user equipment and all $q \cdot (N - 1)$ relays. When a relay $R_i^j$ ($2 \le i \le N$) receives $[s, P_1]$, it generates a parity part $P_i$ by using the encoder in accordance with the source node. Then, the relay generates an STC signal format $[s, P_i]^j$ by using the sequence $[s, P_i]$. Afterwards, it sends the encoded sequence $[s, P_i]^j$ to the user equipment by using the frequency band $f_i$.

When the user equipment receives multiple signals from the relays and the source, it performs an STC decoding process according to the received signals from the same frequency band $f_i$ and recovers the sequence $[s, P_i]$. This decoding process is performed for all the available frequency bands at the user equipment. In the next step, the user equipment combines $[s, P_1], [s, P_2], \ldots,$ and $[s, P_N]$ to regenerate the mother code or a higher rate code, and it applies the iterative method to decode the original information. Even though some of the paths might be severely interfered by environments, the information could still be recovered due to the self-decoded characteristic of the rate compatible code in the system.

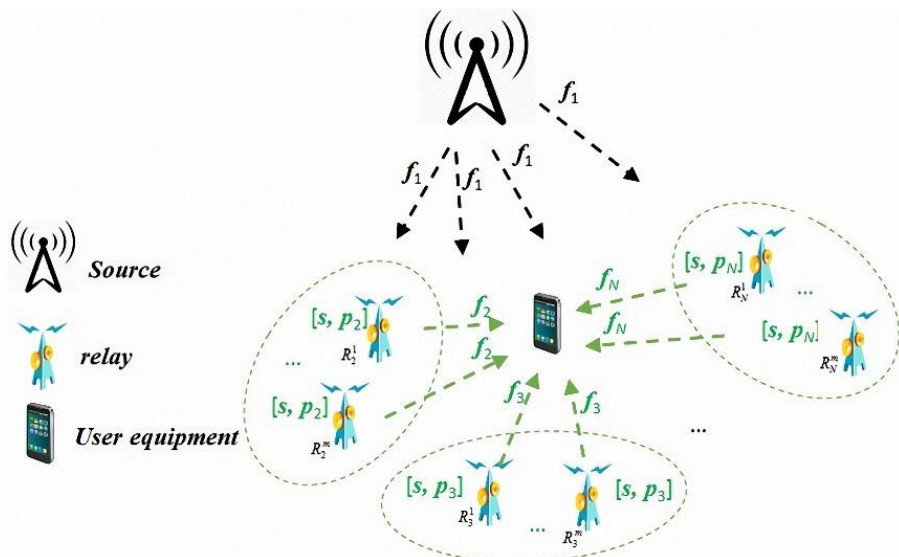

**Figure 1.** The concept of the relay-based cooperative network [16].

In [16] the authors proposed a grouping method named the "all frequency allocated grouping method". The STC can be performed for user equipment in the case of receiving both signals coming from the STC pair stations of the same group. In Figure 1, if there are $(N-1)$ frequencies available for the terrestrial stations(relays), all the stations are divided into $(N-1)$ groups. Each group is assumed to consist of two stations and both two stations transmit signals in the same frequency. Regarding the environment, the channel propagation path loss and fading between each station and the user will be taken into account in this paper. The propagation path loss $L_{ij}$ is defined as the power law of the distance between the $i$-th station and the $j$-th user.

$$L_{ij} = d_{ij}^{-\gamma} \tag{1}$$

where $d_{ij}$ denotes the distance between the $i$-th station and the $j$-th user, and $\gamma$ is the path loss exponent, which is equivalent to 2 in this paper. The fading $X_{ij}$, is assumed to be an independent identical (i.i.d.) Rayleigh random variable with a unit variance. Combining the propagation path loss and the fading, the channel link gain $h_{ij}$ can be represented as (2).

$$\left|h_{ij}\right|^2 = d_{ij}^{-\gamma}\left|X_{ij}\right|^2 \tag{2}$$

when the user receives the STC signals coming from the same group, the expectation value of *SNR* after STC decoding can be shown as (3)

$$\mathrm{E}[SNR] = \frac{P_r}{N_0}\left(\left|h_{2i-1}\right|^2 + \left|h_{2i}\right|^2\right) = \frac{P_r}{N_0}\left(d_{2i-1}^{-\gamma} + d_{2i}^{-\gamma}\right) \tag{3}$$

where $P_r$ is the transmitting power spectral density of each relay and $N_0$ is the noise power spectral density. The maximum ratio combining (MRC) method can be applied after receiving signals coming from each group. Then the expectation value of *SNR* after MRC can be expressed as (4)

$$\mathrm{E}[SNR] = \frac{P_r}{N_0}\left(\left|h_1\right|^2 + \left|h_2\right|^2 + \ldots + \left|h_{2(N-1)}\right|^2\right) = \frac{P_r}{N_0}\left(d_1^{-\gamma} + d_2^{-\gamma} + \ldots + d_{2(N-1)}^{-\gamma}\right) \tag{4}$$

and the channel capacity of a user can be shown as (5) [18]

$$
\begin{aligned}
\mathrm{C} &= \tfrac{1}{N-1} E[\log(1 + SNR)] \approx \tfrac{1}{N-1} \log(1 + E[SNR]) \\
&= \tfrac{1}{N-1} \log\left[1 + \tfrac{P_r}{N_0}\left(d_1^{-\gamma} + d_2^{-\gamma} + \ldots + d_{2(N-1)}^{-\gamma}\right)\right]
\end{aligned}
\tag{5}
$$

Here the coefficient $1/(N-1)$ is a time factor that a user needed to decode the signals coming from $(N-1)$ groups/frequencies. If it is a broadcasting system, no matter how many users are in the system, the channel capacity can be described as (5). However, if it is a service by demand system, different data types are sent to different users. When the number of users $m$ increases, as long as $m \leq (N-1)$, the station in different groups takes turns to transmit data to different users simultaneously. However, once $m > (N-1)$, it will not transmit data from the $(N-1)$ group to $m$ different users simultaneously. The channel capacity could be shown as (6)

$$
\mathrm{C} = \frac{1}{\left(1 + floor\left(\frac{m}{N-1}\right)\right)(N-1)} \log\left[1 + \frac{P_r}{N_0}\left(d_1^{-\gamma} + d_2^{-\gamma} + \ldots + d_{2(N-1)}^{-\gamma}\right)\right]
\tag{6}
$$

The $(1 + floor(m/(N-1)))$ term represents that the system cannot serve the $N$-th user until the data of the previous $(N-1)$ user are sent. The more users in the system, the more waiting time will be needed for the system to serve each user.

There are several problems that will degrade the performance of the system: (1) Receiving and synchronized time for the user equipment will expand whereas the number of the grouping relays increase. (2) The further the station is from the user equipment, the more transmitted power should be transmitted (3) Multicast service or services on demand will spend more time for users to wait. In order to solve the above problems. In [17], we proposed a grouping method called the "pilot-based grouping method". The system model of the pilot-based grouping method is shown in Figure 2.

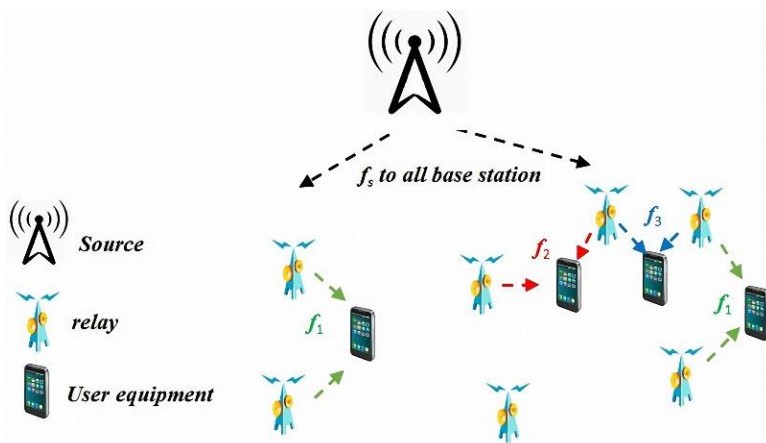

**Figure 2.** The illustration of the pilot-based grouping method [17].

In [17], every terrestrial station transmits signals in the frequency band $(f_1, f_2 \ldots, f_{N-1})$. Each user should be served by the best two relay stations to form an STC pair. Every station will transmit a predefined pilot signal in the centre frequency to identify itself. Then the user in the communication system will detect the pilot channel and report the best two pilot signals to the system. We assumed that the best relay would allocate an available channel to the user, and the second-best relay will follow the same channel chosen by the best relay in order to form an STC pair. Every station will broadcast a predefined pilot signal and channel information in the centre frequency to identify itself. After detecting the pilot signal and recognizing the channel information from each station, the user will allocate one of the same available channels from the best relay and the second relay to form an STC pair.

The basic concept of the pilot-based grouping method is that every user should be served by the best two relays to offer STC, and the same frequency cannot be allocated to different users at the same time. The user will receive the STC signals of the best two relays, and the expected value of SNR can be represented as (7). The channel capacity is given in (8).

$$E[SNR] = \frac{P_r}{N-1} \frac{1}{N_0} \left( |h_1|^2 + |h_2|^2 \right) = \frac{P_r}{(N-1) \, N_0} \left( d_1^{-\gamma} + d_2^{-\gamma} \right) \tag{7}$$

$$\begin{aligned} C &= E[\log(1 + SNR)] \approx \log(1 + E[SNR]) \\ &= \log[1 + \frac{P_r}{(N-1) \cdot N_0}(d_1^{-\gamma} + d_2^{-\gamma})] \end{aligned} \tag{8}$$

In [17], the pilot-based grouping method will first choose the empty channel from its best relay, and the grouping would be successful if the same channel of the second relay is also available. Although the channel capacity of the pilot-based grouping method is better than the results in [16], this procedure prior to the free channel of the best relay will limit the grouping success probability. In order to improve this problem, we proposed a modified pilot-based grouping method in this paper.

### 2.2. Modified Pilot-Based Grouping Method

In [17], the user will find out the best two relays based on the pilot signals and search both of the available channels from these two relays in order to implement the STC technique to overcome the effect of channel fading. However, there is one problem in the channel selection method. The user looks for an available channel from the best relay first and then checks whether the same channel is available in the second-best relay. If the same channel in the second-best relay is unavailable, the grouping will fail even if they still have the same available channels both in the best and the second-best relays.

Instead of making the channels allocation decisions only based on the best relay, the grouping algorithm is modified to assign one of the two available channels based on its best two relays. In this manner, the best two relays have to communicate with each other before allocating the channels. Choosing one of the commonly available channels can surely increase the grouping successful rate defined as (9).

$$\text{The grouping successful rate} = \frac{\text{the number of users who successfully allocated the STC channel}}{\text{total number of users}} \tag{9}$$

For the proposed analysis model, the relays are located in a two-dimensional square area on the x and y axis and the separation of one station to the neighbouring station is one unit on the x and y axis (Figure 3), respectively.

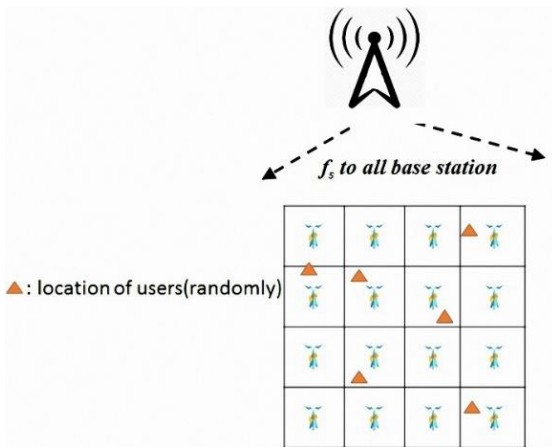

**Figure 3.** Locations of terrestrial station and users.

For the modified pilot-based grouping methods in this paper, there are several initial algorithms introduced as follows (Algorithm 1).

---

**Algorithm 1:** Initialization

---

I.1.  The number of relay $W$ is $2^{2n}$ ($n = 1, 2, 3, \dots$ ).
I.2.  The number of available channels for each station $f_n$ is $W/2$.
I.3.  The stations are uniformly distributed in a square region (length $\times$ width $= 2^n \times 2^n$) at positions ($i + 0.5, j + 0.5$) where $i, j = 0, 1, 2 \dots 2^n - 1$.
I.4.  The total power for each station is $P_r$.
I.5.  All the channels in every station are free to allocate.
I.6.  The number of total users is $M$.
I.7.  The counter for the incoming user $m$ is set to 0.
I.8.  The counter for the number of the grouping success events $n_G$ is set to 0.
I.9.  The counter for the number of the grouping fail event $n_F$ is set to 0.
I.10. The channel capacity for this system $c$ is set to 0.

---

Then, the modified pilot-based method can be described as the following procedure (Algorithm 2).

---

**Algorithm 2:** Modified pilot-based grouping procedure

---

A.1.  Start the procedure
A.2.  If $m \leq M$,

    A.2.1.  $m = m + 1$.
    A.2.2.  The $m$-th user randomly arises in the square region.
    A.2.3.  The $m$-th user chooses the best and second-best pilot signals transmitted from the terrestrial stations, and the best two signals come from $R_{m,1}$, and $R_{m,2}$, respectively.
    A.2.4.  If the number of the common available channels from $R_{m,1}$ and $R_{m,2}$ is larger than zero,

        A.2.4.1. Randomly assigns one of the common available channels to user m.
        A.2.4.2. $n_G = n_G + 1$.
        A.2.4.3. $c = c + 1$.

    A.2.5.  Else

        A.2.5.1  $n_F = n_F + 1$.

    A.2.6.  Go to step A.2.

A.3.  Else

    A.3.1.  Calculate the grouping success rate, $P_G$,

$$P_G = n_G/(n_G + n_F)$$

A.4.  End the procedure.

---

In this paper, there are a total of 16 relays uniformly deployed in the square region and the number of channels in each relay is 8. The transmitting power spectral density $P_r$ in each relay is set to be 10 W/Hz. The environment between each relay and the user will introduce propagation path loss and fading. All the background noises in relays and users are assumed to be an independent identical (i.i.d.) complex Gaussian random variable with a common variance, $N_0$. In this paper, the $N_0$ is assumed to be 1 W/Hz. The path loss is assumed to be the power law of the distance between the relays and the users and is assumed to be 2 for the path loss exponent.

The comparison simulation results between the modified pilot-based grouping method and pilot-based grouping method in [17] are illustrated in Figures 4 and 5. The simulation result in Figure 4 shows that the grouping successful rate of the modified pilot-based grouping method outperforms the pilot-based grouping method in terms of the grouping successful rate, because of the relatively flexible grouping strategy. As the number of users

increases, the grouping success rate of the modified pilot-based grouping method is getting much better than that of the pilot-based grouping method in [17].

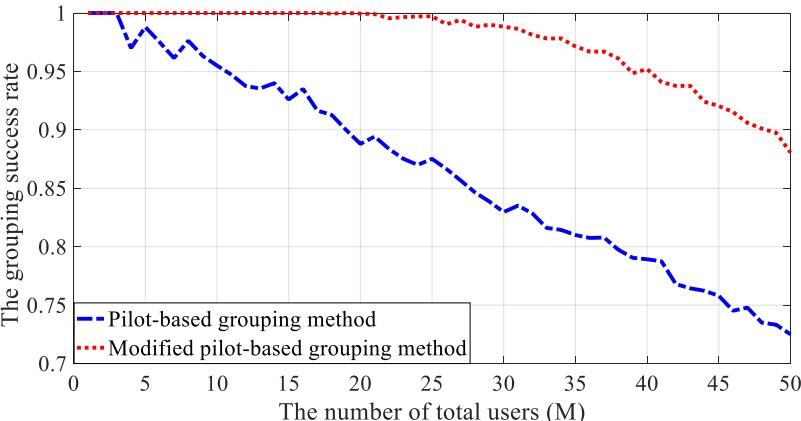

**Figure 4.** The comparison of the grouping successful rate between the modified pilot-based grouping method and the pilot-based grouping method.

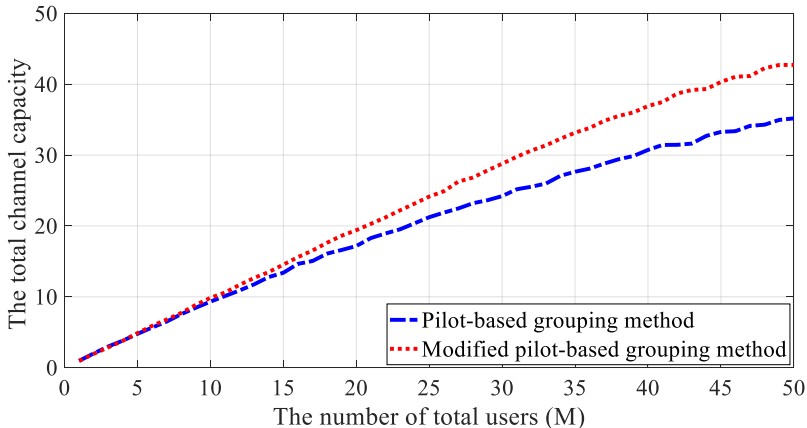

**Figure 5.** The comparison of the total system channel capacity between the modified pilot-based grouping method and the pilot-based grouping method.

Figure 5 shows the comparison of the total system channel capacity between the modified and original pilot-based grouping method in [17]. As a result of the advantage of the grouping successful rate, the modified pilot-based grouping method can be used to increase the total system channel capacity.

### 2.3. Single Relay Allowed Modified Pilot-Based Grouping Method

This method is for a special situation where the user is quite close to the best relay station but fails to group due to the lack of available channel in the second best relay. The user might have a good channel quality to make a transmission with his best relay even without the space time coding gain. In this kind of case, allocating a single channel from the best relay to serve the user should be a good strategy. For the Alamouti space time code in [19], the receivers can decode the signal, whereas only one transmitter's signal is received.

This idea mentioned above indicates that a single relay allowance added in the grouping algorithm would improve the capacity of the system. In this algorithm, the user will choose the available channels both from its best and second-best relay firstly. If there is a lack of commonly available channels from the best and second-best relay, the user will only be assigned the available channel from its best relay. The single relay-allowed modified pilot-based grouping method is described as follows (Algorithm 3).

---

**Algorithm 3:** Single relay allowed modified pilot-based grouping

---

B.1.   Start the procedure

B.2.   If $m \le M$,

    B.2.1.   $m = m + 1$.

    B.2.2.   The $m$-th user randomly arises in the square region.

    B.2.3.   The $m$-th user chooses the best and second-best pilot signals transmitted from the terrestrial stations, and the best two signals come from $R_{m,1}$, and $R_{m,2}$, respectively.

    B.2.4.   If the number of the commonly available channels from $R_{m,1}$ and $R_{m,2}$ is larger than zero,

        B.2.4.1.   Randomly assign one of the common available channels to user $m$.

        B.2.4.2.   $n_G = n_G + 1$.

        B.2.4.3.   $c = c + 1$.

    B.2.5.   Else

        B.2.5.1.   $n_F = n_F + 1$.

        B.2.5.2.   If the number of the available channels from $R_{m,1}$ is larger than zero,

            B.2.5.2.1.   Randomly assigns one of the available channels from $R_{m,1}$ to user $m$.

            B.2.5.2.2.   $c = c + 1$.

    B.2.6.   Go to step B.2.

B.3.   Else

    B.3.1.   Calculate the grouping success rate, $P_G$,

        $P_G = n_G / (n_G + n_F)$

B.4.   End the procedure.

---

In Figure 6, the total channel capacity of both the pilot-based and the modified pilot-based grouping methods can be increased by using single relay allowed strategy. If we look for more details of the simulation result, the increment is more obvious for the pilot-based grouping method than for the modified pilot-based grouping method. One explanation to this result is that the system only benefits from the single relay allowed strategy in case of grouping failure. Since the grouping successful rate of the modified pilot-based grouping method is much better than the original one, it is not surprising that the effect of the single relay-allowed strategy improves channel capacity more than the algorithm with the lower grouping success rate.

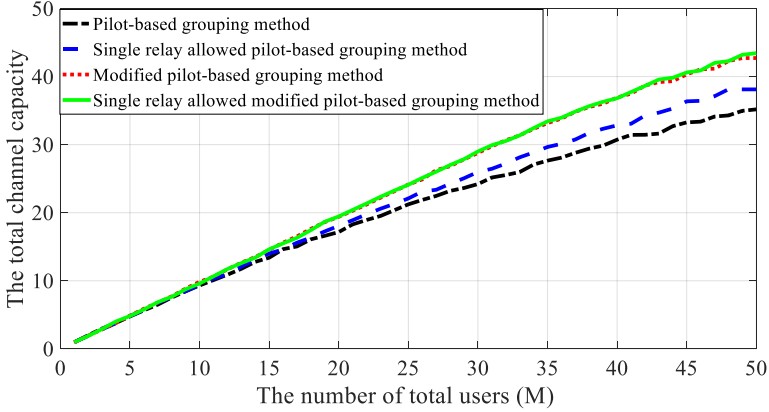

**Figure 6.** The comparison of the total system channel capacity of the pilot-based, single relay allowed pilot-based, modified pilot-based, and single relay-allowed modified pilot-based grouping methods.

## 3. Grouping Method in Multi-Service Environment

MBMS will play an important role in the future. Different services require different transmission bandwidths; for example, video services demand more bandwidth than

audio services. In this paper, the multiple service environment is taken into consideration in a single relay-allowed modified pilot-based grouping method. Due to the different requirements with single service, the initialization of the simulation is modified for a multiple-services environment as follows (Algorithm 4).

---

**Algorithm 4:** Initialization for multiple-services environment

---

J.1.  The number of relay $N$ is $2^{2n}$ ($n$ = 1, 2, 3, … ).
J.2.  The number of available channels for each station $f_n$ is $N/2$.
J.3.  The stations are uniformly distributed in a square region (length × width = $2^n \times 2^n$) at positions ($i$ + 0.5, $j$ + 0.5) where $i, j$ = 0, 1, 2 … $2^n - 1$.
J.4.  The total power for each station is $P_r$.
J.5.  All the channels in every station are free to allocate.
J.6.  The number of total users is $M$, and each user will choose one of $K$ type services according to his requirement.
J.7.  The counter for the incoming user $m$ is set to 0.
J.8.  The counter for the number of the grouping success events $n_G$ is set to 0.
J.9.  The counter for the number of the grouping fail event $n_F$ is set to 0.
J.10.  The channel capacity for this system $c$ is set to 0.
J.11.  Set the occurrence probability $p_S$ and channel requirement for each of $K$ services. Every user will choose one of the services randomly before it starts the grouping procedure.

---

In this paper, three kinds of services are applied: audio, video, and HD-video, assuming the number of channel requirements for these services are 1, 2, and 4, respectively. The probabilities of the different requesting services for users are set to different values. The effect for different service probabilities is also taken into account in our simulation.

The algorithm for the single relay allowed modified pilot-based grouping method in a multiple services environment is described as below (Algorithm 5).

---

**Algorithm 5:** Single relay allowed grouping applying in multi-services environment

---

C.1.  Start the procedure
C.2.  If $m \leq M$,

    C.2.1.  $m$ = $m$ + 1.
    C.2.2.  The $m$-th user chooses one of the $K$ services and randomly arises in the square region.
    C.2.3.  The $m$-th user chooses the best and second-best pilot signals transmitted from the terrestrial stations, and the best two signals come from $R_{m,1}$ and $R_{m,2}$, respectively.
    C.2.4.  If the number of the common available channels from $R_{m,1}$ and $R_{m,2}$ is larger than the $m$-th user's requirement,

        C.2.4.1.  Randomly assign common available channels to user $m$ according to its requirement.
        C.2.4.2.  $n_G$ = $n_G$ + 1.
        C.2.4.3.  $c$ = $c$ + 1.

    C.2.5.  Else

        C.2.5.1.  $n_F$ = $n_F$ + 1.
        C.2.5.2.  If the number of the available channels from $R_{m,1}$ is larger than the $m$-th user's requirement,
        C.2.5.3.  Randomly assign available channels from $R_{m,1}$ to user $m$ according to its requirement.
        C.2.5.4.  $c$ = $c$ + 1.

    C.2.6.  Go to step B.2.

C.3.  Else

    C.3.1.  Calculate the grouping success rate, $P_G$,

$$P_G = n_G/(n_G + n_F)$$

C.4.  End the procedure.

---

Figures 7–9 show the grouping successful rate in the different scenarios. All the results show a trend that the grouping success rate grows, whereas the probability of the lower QoS (quality of service) requirement increases. In contrast, in the higher QoS demands, the grouping success rate will be decayed. Figures 10–12 show the total channel capacity in the different scenarios. When the higher QoS service requirement accounts for the major proportion, the system total capacity will be higher than the lower QoS service requirement. In addition, whereas the number of users grows, the total channel capacity will rise to the upper bound more quickly for a higher QoS service requirement. In short, although the grouping success rate is getting higher, the service for audio is less than that for video and HD video, and the average channel capacity for the higher QoS requirement raises more sharply than that for the lower QoS requirement.

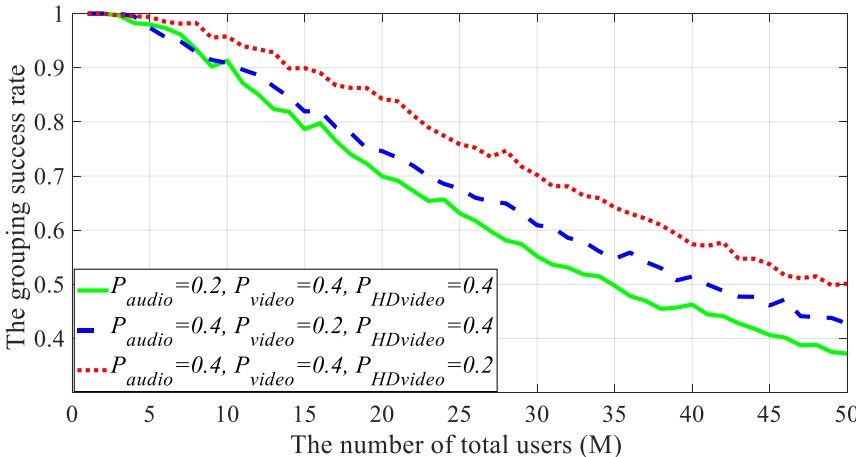

**Figure 7.** The grouping successful rate of the single relay-allowed modified pilot-based grouping method in scenario 1.

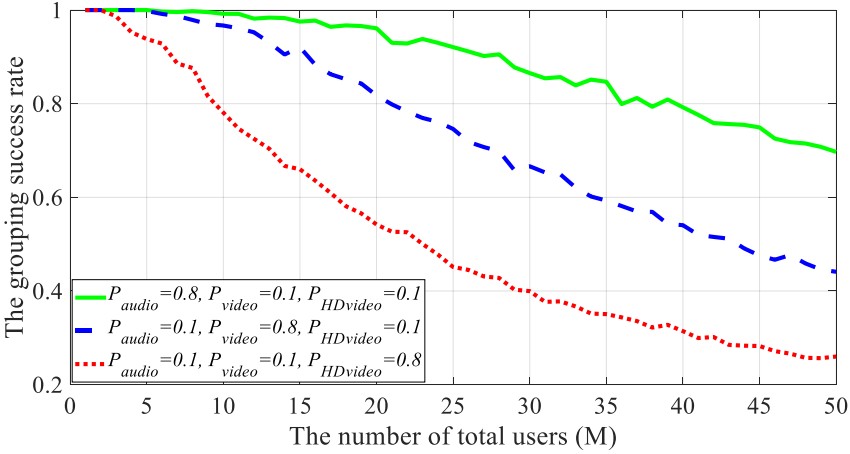

**Figure 8.** The grouping successful rate of the single relay-allowed modified pilot-based grouping method in scenario 2.

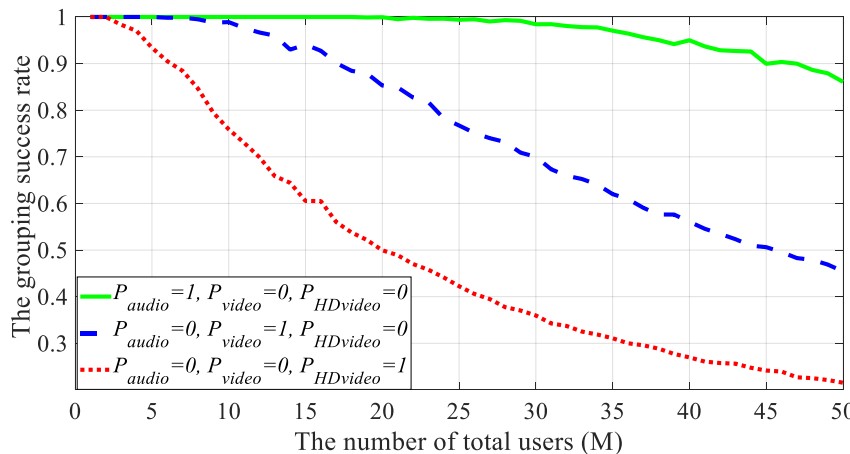

**Figure 9.** The grouping successful rate of the single relay-allowed modified pilot-based grouping method in scenario 3.

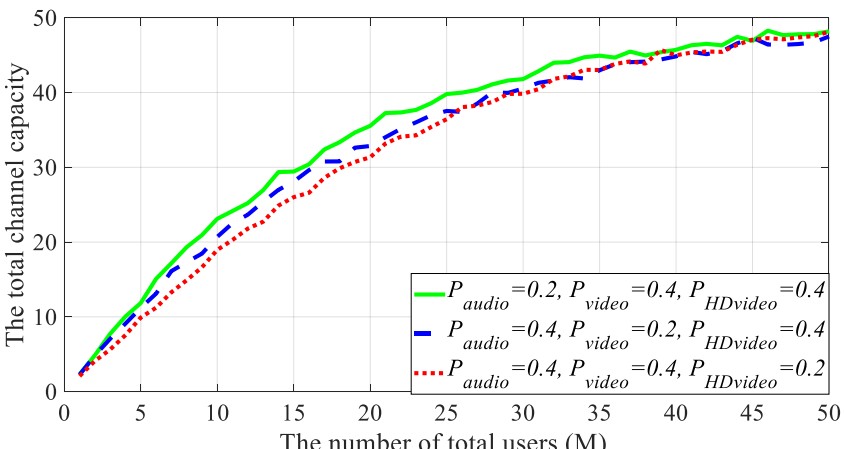

**Figure 10.** The total system channel capacity of the single relay-allowed modified pilot-based grouping method in scenario 1.

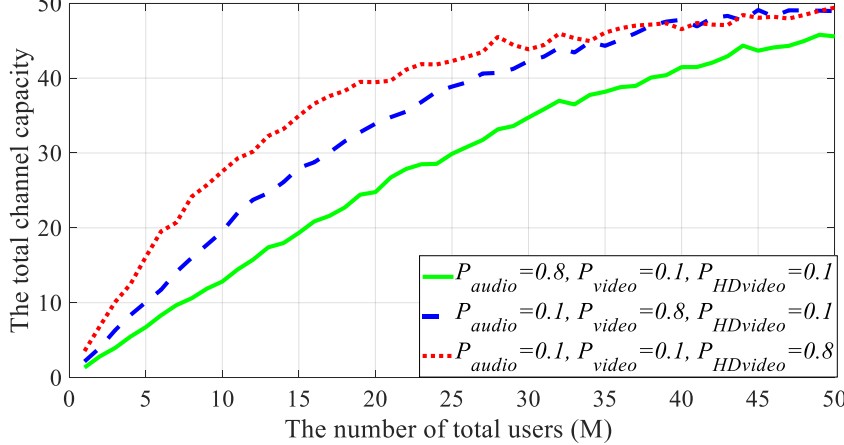

**Figure 11.** The total system channel capacity of the single relay-allowed modified pilot-based grouping method in scenario 2.

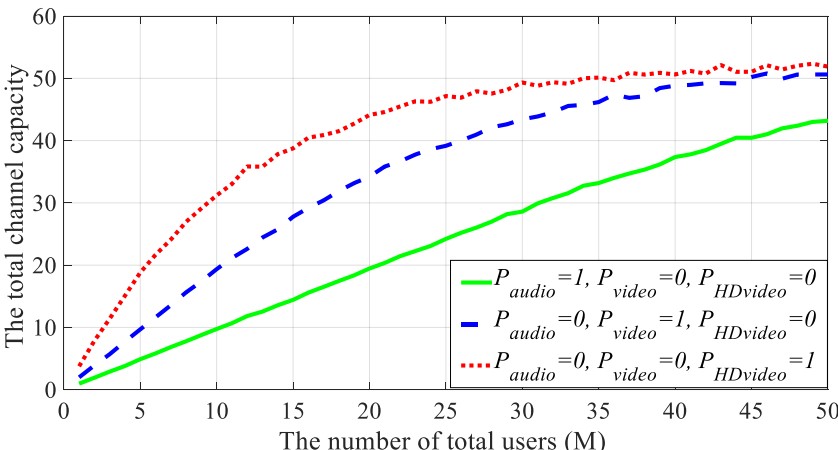

**Figure 12.** The total system channel capacity of the single relay-allowed modified pilot-based grouping method in scenario 3.

## 4. Conclusions

The grouping methods can provide strategies for relay-based cooperative networks to get better channel capacity. In this paper, the modified pilot-based grouping method is proposed to get better total system capacity. The single relay strategy with the modified grouping method also improves the channel capacity and grouping success rate. In addition, because MBMS requires different amounts of content channels to attract users, we also analyzed different grouping scenarios for multi-services. The proposed grouping methods can provide for the satellite-terrestrial communication system to implement the STC technique to increase the grouping success probability. The strategy analysis for multi-services can also be used to implement in MBMS.

Going forward, the proposed grouping method can also be applied in other relay-based cooperative networks, such as Wi-Fi system, ad hoc sensor networks, and IoT networks. Finally, even though our proposed grouping methods can increase the total channel capacity, the mobility issue is worth studying further. For a moving user, the best two relays will be changed, and an error might occur for STC decoding. In the future work, there might be another grouping strategy for moving users.

**Author Contributions:** J.-S.L., J.-H.W., C.-H.C. and C.-E.W. contributed equally to the literature review of pilot-based grouping method, then developed the first discussions about the different parameters of the proposed methodology and conducted the simulations and analysis of results. C.-E.W. guided the investigation and supervised this work. J.-S.L. gave suggestions and guidance for the research. All authors have read and agreed to the published version of the manuscript.

**Funding:** This research received no external funding.

**Data Availability Statement:** The data used to support the finding of this study are included within the article.

**Conflicts of Interest:** The authors declare no conflict of interest.

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
