# Peer review of "Performance on Multiple Pilot-Based Grouping Methods in Satellite-Terrestrial Cooperative Wireless Networks"

_electronics, doi:10.3390/electronics11030430_

Round 1

Reviewer 1 Report

In this paper, the authors present a modified grouping method used as an efficient transmitting strategy in a spatial diversity system. The modified grouping method is compared to others showing a performance improvement in relay-based cooperative networks.

In my opinion, the topic of this work fits the scope of the journal, and it is of scientific interest.

Nonetheless, I have some important concerns that I would like the authors clarify before considering the paper for publication. These concerns are detailed below:

Comments about the writing:

From the viewpoint of this reviewer, the paper needs a thorough review of the writing because there are several typos that hinders the reader from understanding the work. Although I would point out some of these typos, I would kindly recommend the authors to perform a deep review of the paper writing:

Line 17: “… than manners aforementioned…” -> “… than the aforementioned manners …”.

Line 27: “In cellular system … “-> “In a cellular system…”.

Line 29: “…authors want…” -> “…authors wanted…”, please be consistent in the use of verb tenses. The authors use the past tense in the sentence before and after this one.

Line 42: “In US…” -> “In the US…”.

Line 47: “… serval study…” -> “… several studies…”.

Line 57: “Fist…” -> “First…”.

Line 58: “The far distance between…, the more transmitted…”, this reviewer thinks that the authors may want to say: “The further is the station from the user equipment, the more transmitted power should be transmitted.”. Nevertheless, try to rephrase in the clearest possible way, please.

Line 68: “… are take into account.” -> “… are taken into account.”.

And so on. Please, make a thorough review of the writing.

Please, improve the quality of the figures, it seems that they are not vector graphics and zooming in does not provide a clear sight of the figure since they are lightly blurred.

Comments about the contents of the paper:

With respect to the title, this reviewer does not clearly see the need to include the term “Satellite-Terrestrial” because that is not clear at all from the explanations and from the figures included in the paper.

Section 2:

Line 81: Please, rephrase “N is the total channel numbers for each relay”. It seems that the authors meant that each relay uses N channels, but, from Figure 1 it seems that each relay transmits only over one frequency channel. Thus, it seems that N is the total number of (frequency) channels.

Line 84: “spilt to m groups”, the authors may want to say:” split into groups of m relays”. Please, it seems that there are N-1 relay groups each one consisting of m relay elements.

In line 106-107, the authors claim that “The channel capacity [16] of a user can be shown as (1)”. Nevertheless, as far as this reviewer understands, equation (1) does not appear in reference [16]. Where does this equation come from? How did the authors get it? Please, clearly explain the terms in that equation. For instance: explain where does “dpq” appear in equation (1). In addition, it seems that variable “m” has changed its meaning (in such equation is the number of served users and not the number of relays in each frequency band). Moreover, it is also misleading the term “N-th” user, since N-1 was used as the number relay groups. Finally, N0 seems to be “noise power spectral density” and not “noise power” attending to the units (W/Hz) given in line 173.

The comments of the above paragraph apply to equation (2). Please, explain how it is obtained and explain the components of the equation.

The different procedures use a notation that is not consistent with the beginning of the section. For instance, user index is ‘m’, while ‘m’ was the number of relays in each relay group in figure 1; ‘N’ is now the total number of relays and not ‘N-1’.

In lines 154-156, the authors claim that “In this manner, the best two relays have to communicate with each other before allocating the channels. Choosing one of the common available channels can surely increase the grouping successful rate defined as (3)”. Considering this statement, it seems that in the method suggested by the authors it is necessary to stablish an additional communication between the two best relays to decide and inform about the channel to be used. How does this communication affect to the capacity of the system in a real-world mobile scenario where the users might be moving from one relay to another relay?

Please, rephrase the sentence in lines 200-202 as a statement. It seems more appropriate that style for a journal.

Section 3:

Please, correct the probabilities that appear associated to the solid-line (black) curve in figures 9 and 12.

Round 2

Reviewer 1 Report

The authors have adequately answered to some of this reviewer’s comments. Nevertheless, there still are some concerns that prevent this reviewer to accept the paper in its present form. These concerns are detailed below:

Please, review again the writing. Among others, this reviewer has found the following typos:

Line 15: “A grouping method… had proposed in our…”.

Line 20: “… networks are proposed   The…”

Line 49: “In view of the foregoing… serval fields…”.

Line 52: “ … in how to category the relays…”. Category is noun, the verb related to putting in groups depending on their type is “categorise”.

Line 107: “Each group is assumed to be consists of two stations…”

The authors claim in Line 107 that each group consists of two stations, but figure 1 plots more than two. In addition, from the definition of (1) and (2), it seems that h_ij is the channel link gain (CLG) between the i-th station and the j-th user, so h_2i-1 and h_2i are, respectively, the CLG between station 2 and users i-1 and i. It seems that the authors wanted to express that h_1i and h_2i (the two stations from the same group that reach user i).

The issue mentioned in the above paragraph affects also to equation (4). More about equation (4): what does h_1 and h2 mean? What does the sum |h1|^2+|h2|^2+…+|h_2(N-1)|^2 mean?

As far as this reviewer understands, if X_ij is a Rayleigh random variable (RRV), then its mean cannot be 0 (except for a trivial case); thus, if X_ij is of unit variance, then E{|X_ij|^2} is not 1. Its variance is one, but not the expected value of its squared value.

Line 130, it seems that the authors meant that the referred term represents that the system cannot serve (and not “can serve”) the N-th user.

Equation (9) may benefit from algebraic notation. In addition, methods like “Pilot-based grouping method” or “Modified pilot-based grouping method”, may also be written in terms of acronyms in the figures 4, 5, and so on. X-axis of those graphs are clearer just with “M” that should be clearly defined in the text.
